# Effect of Carbohydrate-Restricted Dietary Pattern on Insulin Treatment Rate, Lipid Metabolism and Nutritional Status in Pregnant Women with Gestational Diabetes in Beijing, China

**DOI:** 10.3390/nu14020359

**Published:** 2022-01-14

**Authors:** Mingxuan Cui, Xuening Li, Chen Yang, Linlin Wang, Lulu Lu, Shilong Zhao, Qianying Guo, Peng Liu

**Affiliations:** 1Department of Clinical Nutrition, Peking University People’s Hospital, Beijing 100044, China; cmxjuicy@bjmu.edu.cn (M.C.); lxn@bjmu.edu.cn (X.L.); LLL163youxiang@163.com (L.L.); zhaoshilongrmyy@163.com (S.Z.); 2Institute of Reproductive and Child Health, National Health Commission Key Laboratory of Reproductive Health, School of Public Health, Peking University, Beijing 100191, China; yangchen@bjmu.edu.cn (C.Y.); linlinwang@bjmu.edu.cn (L.W.)

**Keywords:** carbohydrate-restricted dietary pattern, gestational diabetes mellitus, insulin treatment rate, lipid metabolism, nutritional status

## Abstract

Carbohydrates play an important role in blood glucose control in pregnant women with GDM. Carbohydrate-restricted dietary (CRD) pattern for gestational diabetes mellitus (GDM) has been widely used in clinics, but the change in insulin utilization rate beyond CRD intervention in GDM remains unclear. The aim of the present study was to explore the application of insulin in pregnancy with GDM, as well as the influence of CRD pattern on lipid metabolism and nutritional state. A retrospective study of 265 women with GDM who delivered in Peking University People’s Hospital from July 2018 to January 2020 was conducted using a questionnaire survey. Women were divided into a CRD group or a control group according to whether they had received CRD intervention during pregnancy. There was no statistically significant difference in the rate of insulin therapy between the two groups (*p* > 0.05), the initial gestational week of the CRD group combined with insulin treatment was significantly higher than that of the control group (*p* < 0.05), and the risk of insulin therapy was positively correlated with fasting plasma glucose (FPG) in early pregnancy (*p* < 0.05). The incidence of abnormal low-density lipoprotein cholesterol levels in the CRD group was significantly lower than that in the control group (*p* < 0.05). There were no significant differences in nutritional indexes between the two groups. The results indicate that CRD intervention may be effective in delaying the use of insulin and improving the blood lipids metabolism during GDM pregnancy, while nutritional status may not be significantly affected under CRD intervention, and a high FPG in early pregnancy with GDM may be a risk factor for combined insulin therapy with CRD intervention.

## 1. Introduction

Gestational diabetes mellitus (GDM) is one of the most common complications of pregnancy, and the number of women diagnosed with GDM is increasing worldwide [1]. GDM is a pregnancy-specific disease that harms both maternal and fetal health. Improper management of blood glucose during pregnancy can lead to premature delivery, dystocia, polyhydramnios, macrosomia, fetal growth restriction, fetal distress, fetal death in utero, neonatal respiratory distress syndrome and neonatal hypoglycemia, as well as other adverse pregnancy outcomes. In terms of long-term effects, women with a history of GDM have a significantly increased risk of long-term obesity, diabetes and cardiovascular diseases, and the offspring of GDM patients also have a significantly increased risk of developing type 2 diabetes, obesity and metabolic syndrome, posing a serious threat to the health of mothers and children [2,3].

The etiology of GDM is complicated and remains unclear. GDM might be associated with genetics; changes in hormone levels during pregnancy, which may lead to insulin resistance [4]; elevated levels of inflammatory cytokines [5,6]; and lifestyle changes, including diet and exercise. 

Dietary therapy is the cornerstone of diabetes treatment. The effectiveness of medical nutritional therapy (MNT) for GDM has been widely confirmed and has been applied in clinical practice. MNT is a general term for nutritional intervention measures for nutrition-related problems under clinical conditions, including individualized nutritional assessment and the diagnosis of patients and the formulation of corresponding nutritional intervention programs [7]. The purpose of MNT intervention for GDM patients is to correct the existing abnormal glucose and lipid metabolism by adjusting the diet structure and maintaining the normal nutritional status so as to promote patients’ blood glucose to reach the standard and reduce the risk of maternal and infant complications [8]. Most studies have confirmed that about 70~85% of GDM patients can control their blood glucose through simple lifestyle changes [9]. To date, the effectiveness of a carbohydrate-restricted dietary (CRD) pattern as an MNT intervention for the glycemic control of GDM has been verified by some studies, but its impact on the drug treatment rate of GDM remains unclear.

Carbohydrate (CHO) is the macronutrient that has the greatest influence on postprandial blood glucose response. MNT for GDM includes dietary guidance on the distribution, type and quantity of CHOs [10,11,12]. It is customary in China to recommend appropriate limits on the total CHO intake of pregnant women throughout the day, while ensuring full nutrition during pregnancy, including adequate intake of protein, fat and micronutrients. A CRD pattern can be generally defined as a dietary pattern with a relatively low CHO intake [13]. According to the diagnosis and therapy guidelines of pregnancy with diabetes mellitus in China, daily CHO intake <50% E throughout pregnancy was regarded as the CRD pattern in this study [14]. Spreading CHOs throughout the day over multiple meals and snacks may help control postprandial blood glucose levels [10]. Thus, it is recommended that GDM patients eat at least five–six meals a day, that is, two–three snacks a day in addition to three main meals in the morning, noon and evening. The CRD pattern used in the present study included a restricted whole-day CHO intake and a limit of CHO intake at each meal. Evidence based on most studies demonstrates that CRD pattern plays a significant role in the management of GDM, which may improve maternal glycemia and pregnancy outcomes, but the results remain controversial mainly because of the varying degrees of CHO restriction [15,16]. A less restrictive nutritional approach may lead to similar and potentially more favorable outcomes and relieve the anxiety caused by the diagnosis and treatment of the disease [16,17].

There is a close relationship between glucose and lipid metabolism in GDM. Since abnormal lipid metabolism can promote the development of GDM, controlling blood lipid levels in the normal range is beneficial to delay the occurrence of metabolic complications during pregnancy [18]. A meta-analysis of 60 studies showed that triglycerides (TGs) and non-high-density lipoprotein cholesterol were significantly higher and high-density lipoprotein cholesterol (HDLC) levels were significantly lower in women with GDM compared to women with normal pregnancy [19]. Dyslipidemia in GDM may also affect fetal growth, and a prospective study confirmed that maternal TG and free fatty acid levels are independent risk factors for higher gestational age [20]. There are several studies about the relationship between dietary patterns and blood lipids, but few focus on the pregnant population, especially individuals with glucose metabolism disorders. Yancy et al. conducted a 24-week trial to study the effect of a low-CHO diet on blood lipids and found that a low-CHO diet reduced serum TG levels and increased HDLC levels more than the low-fat diet among overweight, hyperlipidemic volunteers [21]. In this study, we intended to determine whether the CRD pattern could help in lipid management in women with GDM.

The CRD pattern might improve the glucolipid metabolism during pregnancy with GDM. Primarily, we aimed to investigate the effect of the CRD pattern on glycemic management in GDM patients by analyzing insulin use. Secondarily, we explored the effect of the CRD pattern on lipid metabolism in GDM patients and assessed the differences in nutritional indicators between patients with and without CRD intervention to observe whether the CRD pattern had an impact on the nutritional status of pregnant women with GDM.

## 2. Materials and Methods

### 2.1. Study Design

We conducted a case–control study to investigate the effect of the CRD pattern on glucose-related metabolic and nutritional states in pregnant women with GDM. A total of 265 women with GDM who delivered in Peking University People’s Hospital from July 2018 to January 2020 were included. 

Eligible pregnant women aged at least 18 years were identified following diagnosis of GDM based on the 75 g oral glucose tolerance test (OGTT) diagnostic criteria published by the International Association of Diabetes and Pregnancy Study Groups (IADPSG) in 2010 [22]. Between 24 and 28 weeks of gestation, level of blood glucose should be lower than 5.1, 10.0 and 8.5 mmol/L (92, 180 and 153 mg/dL) before and 1 h and 2 h after glucose administration, respectively. Any blood glucose level that meets or exceeds these criteria is diagnosed as GDM. Women with prior history of diabetes, dyslipidemia, serious systemic diseases, severe postpartum complications, abnormal Apgar score in neonates and less than 30 min of activity per day or more than 2 h per day were excluded. Study protocol was approved by the research ethics committee of Peking University People’s Hospital, and informed consent was taken from all participants.

General data were collected using a questionnaire survey, and GDM women were asked whether they were willing to participate in the research-related questionnaire survey during hospital delivery. The questionnaire included age, height, pre-pregnancy weight, prenatal weight, pregnancy weight gain, clinical nutrition experience during pregnancy and specific dietary intervention programs. In addition, data of maternal history, pregnancy history, medical advice during pregnancy, prenatal abdominal circumference, delivery method, birth weight and length of newborn were collected by consulting medical records. 

### 2.2. Intervention

All participants were divided into either a CRD group or control group according to whether they had received a CRD plan from a dietician during pregnancy. According to the questionnaire, they were grouped after answering two questions—“Have you received dietary guidance related to the CRD pattern from a dietician” and “Implementation of the CRD program (50% or less)”. GDM pregnant women who “Received CRD pattern instruction” and “Executed more than 50%” were classified as the CRD group, while those who “Did not receive CRD pattern instruction” and those who received CRD pattern instruction but executed less than 50% were classified as the control group.

Women in the CRD group were required to follow a dietary plan developed by a dietician during their visit to a nutrition clinic at pregnancy. At the time of the first visit, the dietitian had taken measurements of the pregnant women with GDM, including height, weight and abdominal circumference, and conducted a dietary survey (24 h dietary review) and individualized MNT program. In general, the meal plans and the exercise plans included in the MNT programs followed the dietary guidelines for pregnant women with GDM in China. 

According to the gestational weeks and individual needs, calorie content of the CRD meal plans were from 1800 to 2200 kcal. Calories were calculated using Henry’s [23] formula for calculating resting energy expenditure (REE), which is based on pre-pregnancy weight and height multiplied by a physical activity level factor. All patients were in their second or third trimester when meeting with a dietician, and an additional 200 kcal were added. According to the recommendation of the Institute of Medicine (IOM) [24], the estimated calorie needs were reduced by 30% if the patient was overweight (pre-pregnancy BMI ≥ 24 kg/m^2^) or had already reached the standard weight gain. All meal plans provided a total of five to seven meals a day. The overall macronutrient distribution of the CRD group was the same, including 45–50 E% CHO, 25–30 E% fat and 20–25 E% protein, according to a recent systematic review of CRD pattern and Chinese guidelines for GDM diagnosis and treatment [13,14]. Meanwhile, the dietary composition of the control group included 60–65 E% CHO, 20–30 E% fat and 15–20 E% protein. Daily energy intakes of the control group were equal to those of the CRD group, ranging from 1800 to 2200 kcal. After calculating each participant’s individual energy needs, light to moderate activity for 20–30 min after three meals was used to make exercise plans. The principles of MNT followed in this study and meal plans, including the distribution of energy, CHO, fat and protein content during the day, in CRD group are shown in Table 1.

Although the meal plans were conducted in home settings, the dietitian followed up every patient who received CRD intervention on a regular basis. The GDM patients visited the clinic once a week during the CRD intervention period, which lasted three to four weeks on average. Meal plans with individualized CHO distribution were handed out at the beginning of each intervention period. In each outpatient follow-up, patients were required to carry a 3-day diet diary. Before that, the participants were provided with models of common food items for estimating food weight. All participants were asked to weigh all foods and pictures of the main meals and of the plate if any leftovers were to be taken so that the dietitian could visually assess how much a patient was eating. All meals contained more than one-third of whole-grain products. The participants were asked not to ingest or drink anything besides the food components of the meal plan. In addition, women who were not consulted promptly from the time of GDM diagnosis (24–28 weeks) to the time of delivery were excluded.

### 2.3. Self-Monitoring of Blood Glucose (SMBG)

Each participant was instructed to carry out SMBG four times a day (before breakfast, 2 h after breakfast, lunch and dinner) with a glucometer. Patients in CRD group recorded the SMBG data in their 3-day food diary as requested by the dietician.

### 2.4. Laboratory Examination

FPG (fasting plasma glucose), lipid profile (TC (total cholesterol), TG, HDLC and LDLC (low-density lipoprotein cholesterol)) and nutrition-related indicators (HB (hemoglobin), ALB (albumin) and TP (total protein)) were measured during the second and third trimesters. FPG of the first trimester was measured at the first inspection. OGTT was performed at 24 to 28 weeks in all participants.

### 2.5. Study Outcomes

Insulin treatment rate as the primary outcome was compared between the two groups, as few studies have previously assessed the use of insulin from GDM patients, especially in China. The initial gestational age of insulin use between the two groups and the risk factors for insulin use in the CRD group were further analyzed.

Levels of serum lipid and nutrition indicators during pregnancy under CRD intervention were analyzed as secondary outcomes. This study explored the changes in lipid metabolism and nutritional status, which were associated with glucose metabolism under CRD pattern of GDM pregnancy.

### 2.6. Sample Size Estimate

The primary clinical endpoint was the difference in insulin utilization rate between intervention group and control group. We used SAS 9.4 to estimate the sample size based on a foreign study on insulin application in pregnant women with GDM under a high-glycemic index (GI) diet and low-GI diet, which showed that the utilization rates of insulin in the two groups were 59% and 29%, respectively [16]. A study size of 264 participants was required (α = 0.05, 1 − β = 0.8).

### 2.7. Statistical Analysis

SPSS 20.0 software was used for data statistical analysis. Kolmogorov–Smirnov was used for normality test. Measurement data conforming to normal distribution were expressed by ($\bar{x}$ ± s), while data that did not conform to normal distribution were represented by median (P25~P75), and a non-parametric test was used for comparison between groups. The count data were expressed by the number of cases and rate, and the comparison between groups was performed using a χ^2^ test, including cases of insulin use and serum lipid parameters. To analyze the initial treatment time of insulin and nutritional status, independent sample T test was used for inter-group comparison. Univariate analysis and binary logistic regression were conducted to analyze the influencing factors of insulin treatment in CRD group. *p* < 0. 05 was considered statistically significant.

## 3. Results

### 3.1. Participants

At the beginning of the study, 353 patients with GDM were invited to fill in the questionnaire to enter the trial; however, 65 participants refused to cooperate, and 23 questionnaires were excluded due to unavailable data. Finally, 265 participants (control group (*n* = 113) and CRD group (*n* = 152)) were included in the analysis (Figure 1).

The baseline characteristics of the participants in the two groups are shown in Table 2. Mean height, median of age, pre-pregnancy weight, BMI, gestational weight gain, antenatal abdominal perimeter, delivery mode and infant birth length were not statistically different between the CRD and control groups. The median of infant birthweight and the proportion of parturient women in the CRD group were significantly higher than those in the control group.

### 3.2. Differences in Insulin Treatment Rate and Risk Factors of Insulin Use

There was no statistically significant difference in insulin application rate between the two groups. When compared with parturients in the control group, the initial gestational weeks of insulin treatment of parturients in the CRD group were significantly later, as shown in Table 3.

Univariate analysis was conducted for the CRD group. Compared with the CRD intervention alone, the women treated with insulin were older, had higher FPG in the early and middle trimesters and had higher glucose levels at 60 min of OGTT, as shown in Table 4.

Logistic regression analysis of risk factors for insulin use in the CRD group was performed, showing that GDM mothers with higher FPG in the first trimester had a higher risk of insulin use during pregnancy (OR = 8.203, 95%CI 2.115 to 31.812; *p* = 0.002). 

### 3.3. Effect of CRD Pattern on Lipid Metabolism 

Lipid levels between the CRD group and the control group were compared by T test and no statistically significant difference was found in both the second and third trimesters (Figure 2).

According to the χ^2^ test, there was a statistically significant difference between the CRD group and the control group in the proportion of abnormal LDLC in the second trimester (Table 5).

### 3.4. Nutrition Status

We conducted a statistical analysis on the nutritional status of pregnant women with GDM in the middle and third trimesters, and we found that there was no statistical difference in TP, ALB or HB between the CRD group and the control group (*p* > 0.05), as shown in Table 6.

## 4. Discussion

Nutrition is critical to the treatment and prevention of GDM for the health of both mother and offspring. In addition, nutritional intervention of GDM may serve as a starting point for healthy dietary pattern transformation during pregnancy, which is conducive to the persistence of a healthy dietary pattern after delivery and has a protective effect on long-term metabolic diseases, such as type 2 diabetes [10]. Close attention to the amount or type of dietary CHO can have important benefits on the pathophysiology of GDM. As a result, most dietary guidelines currently recommend either to restrict CHO intake or to eat low-GI CHOs instead of those that are more quickly digested [25].

Studies have confirmed that the quantity and type of dietary CHO may have an influence on maternal glucose, and women with GDM were recently advised to limit total intake or choose low-GI CHO by nutritional recommendations [26]. Related studies have set energy restrictions through changing the type or reducing the quantity of CHOs with or without increasing physical activity to slow or reduce weight gain during GDM pregnancy [27]. Although several trials [28] that assessed dietary regimens focusing on CHOs in GDM pregnant women did not show consistency in perinatal outcomes, the meta-analysis by Wan et al. [29], who focused on all Chinese-language studies, showed that CHO-modified dietary intervention strategies were associated with improved glycemic control, as well as maternal and infant outcomes in ethnic Chinese women with GDM. 

The relevant guidelines and a number of studies have confirmed that MNT is the initial treatment for GDM, and drug therapy should be performed after MNT has a poor intervention effect [11,12]. Currently, most studies focus on the levels of glucose-related indicators during pregnancy and perinatal outcomes, and they explore the effectiveness of CRD pattern in the blood glucose management of GDM [11]. The use of insulin during pregnancy as an indicator of the effectiveness of CRD intervention was observed in the present study. We explored whether GDM women who had received CRD intervention during pregnancy had a lower insulin treatment rate. Although no significant difference was found in the rate of insulin treatment between the two groups, the onset of insulin treatment in the CRD group was significantly later than that in the control group. A review of the dietary interventions for GDM showed that a low-GI diet characterized by the intake of high-quality complex CHO can reduce insulin utilization and the risk of macrosomia [30]. The use of insulin may be affected by a variety of factors in clinical practice, and the compliance with a CRD regimen in GDM patients also affects the effect of nutritional intervention. The results of this study confirmed the treatment effectiveness for GDM from the perspective that CRD intervention delayed the timing of insulin therapy.

In this study, the risk factors of insulin therapy in the CRD group were preliminarily explored, and it was found that the higher the FPG in the first trimester, the higher the risk of insulin use during GDM pregnancy. Previous studies have shown that women with GDM who fail to achieve good glycemic control through lifestyle changes alone may have a range of characteristics, including baseline characteristics, such as BMI, age and diagnostic parameters of diabetes [31]. Similarly, we found that gestational age, early pregnancy FPG and the level of OGTT 60 min were significantly higher in the CRD combined with insulin group than those in the non-insulin group by univariate analysis. Some studies have speculated that women with GDM who require MNT+ insulin therapy may require more rigorous risk screening and early diagnosis as a special type of GDM [22]. Barnes RA et al. found that gestational age > 30 years, early diagnosis of GDM (<24 weeks of gestation) and fasting venous glucose level (≥5.3 mmol/L) were independent predictors of insulin therapy for GDM [32], which is consistent with the present study. Therefore, early risk control and early identification of GDM risk factors are crucial to improve the effectiveness of GDM intervention.

The glycolipid metabolism of GDM has a certain interaction. In order to maintain the normal development of the fetus and their own energy metabolism, the pregnant women’s ability to absorb fat is significantly enhanced, forming physiological hyperlipidemia [18]. Studies have found that lipid metabolism disorders begin in the early stages of the disease and play an important role in the development of metabolic diseases during pregnancy [33]. We found that CRD intervention can significantly reduce the incidence of an abnormal LDLC level in the middle of GDM pregnancy. This study preliminarily confirmed that CRD pattern plays a certain role in the improvement of the LDLC level of GDM in the second trimester, which is consistent with a series of studies that found that specific forms of MNT intervention can improve lipid metabolism, control weight gain during pregnancy and improve pregnancy outcomes [34].

Adequate nutrition during pregnancy is important for the health of both mother and child. Recent studies showed that, under the conditions of CHO restriction, fuel sources shift from glucose and fatty acids to fatty acids and ketones, and CRD pattern leads to appetite reduction and weight loss [35]. In order to observe whether CRD pattern causes malnutrition in pregnant women with GDM in the study, common nutrition-related indicators of pregnant women in the CRD group and the control group were compared, and no significant statistical differences were found between any indicators. It was preliminarily confirmed that a CRD pattern with 45%–50% E CHO has no risk of malnutrition.

## 5. Conclusions

In summary, the CRD pattern can effectively delay the use of insulin during pregnancy with GDM, which confirms the effectiveness of CRD pattern in blood glucose management from this perspective. High FPG in early pregnancy may be the most important risk factor for GDM in pregnancy treated with a CRD pattern combined with insulin. CRD pattern may have a certain regulation effect on lipid metabolism during GDM pregnancy, and no obvious adverse effect was found on the nutritional status of pregnant women. Further study about the effect of CRD pattern on other metabolisms of GDM pregnant women is urgently needed.

## Figures and Tables

**Figure 1 nutrients-14-00359-f001:**
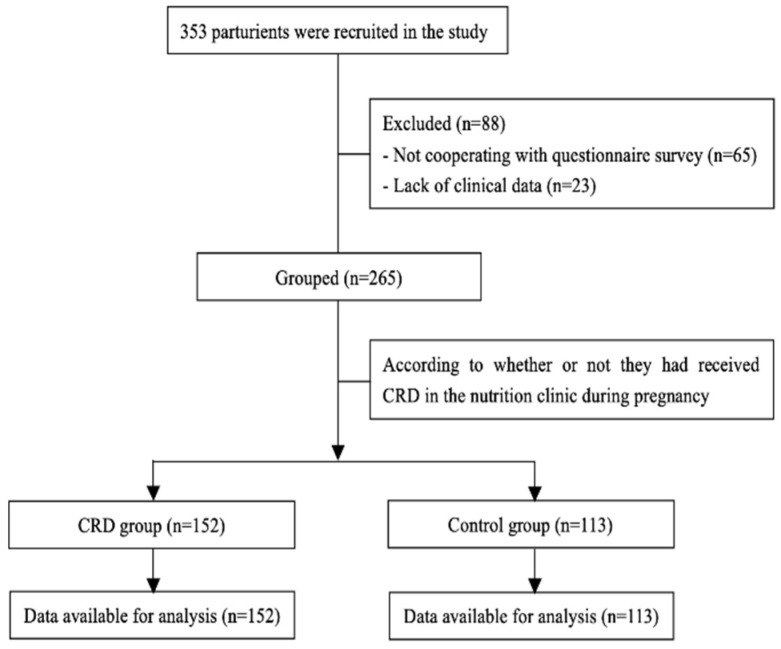
Flow of participants in the study.

**Figure 2 nutrients-14-00359-f002:**
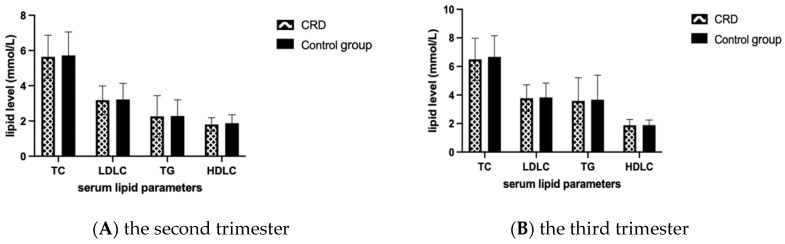
Comparison of lipid-related indexes between the two groups. Mean and SD of TC, TG, HDLC and LDLC levels in the second and third trimester in each group are presented in panel (**A**) and (**B**), respectively. TC, total cholesterol; TG, triglyceride; HDLC, high-density lipoprotein cholesterol; LDLC, low-density lipoprotein cholesterol.

**Table 1 nutrients-14-00359-t001:** Nutrient structure and diet compositions of CRD pattern.

Nutrient ^1^	Energy Distribution ^2^	CHO Distribution ^3^
CHO 45–50 E% Fiber 25–30 g No sugar added Protein 20–25 E% Fat 25–30 E% SFA < 7 E% MUFA > 1/3 E% from fat	Breakfast: 20% Morning snack: 0–15% Lunch: 25–30% Afternoon snack: 10–15% Dinner: 25–30% Late-night snack: 10–15%	Breakfast: 20% Morning snack:0–15% Lunch: 15–20% Afternoon snack: 15–20% Dinner: 15–20% Late-night snack: 15–20%

^1^ SFA, saturated fatty acid; MUFA, monounsaturated fatty acid; ^2^ energy as a percentage of total daily calories; ^3^ CHO as a percentage of total daily CHO intake.

**Table 2 nutrients-14-00359-t002:** Baseline characteristics.

Characteristic	CRD Group (*n* = 152)	Control Group (*n* = 113)
Age at delivery (years) ^#^	34.5 (31.0, 38.0)	34.0 (31.0, 37.0)
Height (cm) *	162.1 ± 5.2	162.0 ± 5.5
Pre-pregnancy weight (kg) ^#^	54.2 (47.1, 61.9)	56.0 (50.0, 63.5)
Pre-pregnancy BMI (kg/m^2^) ^#^	20.3 (16.9, 23.0)	21.3 (18.3, 24.0)
Gestational weight gain (kg) ^#^	11.8 (9.0, 15.0)	12.0 (10.0, 15.0)
Antenatal abdominal perimeter (cm) ^#^	103.0 (97.0, 106.0)	102.0 (99.0, 108.0)
Infant birthweight (g) ^#^	3470.0 (3192.5, 3715.0)	3310.0 (3045.0, 3660.0)
Infant birth length (cm) ^#^	50.0 (49.0, 51.0)	50.0 (49.0, 51.0)
Parity (*n*(%)) ^+^		
≥1	96 (63.2%)	63 (55.8%)
0	56 (36.8%)	50 (44.2%)
Delivery mode (*n*(%)) ^+^		
Natural birth	79 (52.0%)	57 (50.4%)
Cesarean delivery	73 (48.0%)	56 (49.6%)

* Data are mean ± SDs. ^#^ Data are median and interquartile ranges. ^+^ Data are numbers and %.

**Table 3 nutrients-14-00359-t003:** Comparison of insulin treatment rate and the initial gestational weeks.

Insulin Use	CRD Group (*n* = 152)	Control Group (*n* = 113)	χ^2^/t	*p*
Number of cases (%)	22 (14.5)	14 (12.4)	0.24	0.624
Initial treatment time (gestational weeks)	33.59 ± 3.45	29.21 ± 4.07	−3.47	0.001

**Table 4 nutrients-14-00359-t004:** Univariate comparison of risk factors for insulin use in CRD group.

	Treated with Insulin (*n* = 24)	No Insulin Treatment (*n* = 128)	χ^2^/t/Z	*p*
Age at delivery (years) ^#^	37.50 (34.53, 39.97)	34.00 (33.36, 34.95)	−2.215	0.027
Pre-pregnancy BMI (kg/m^2^) ^#^	21.62 (20.50, 24.10)	20.27 (19.43, 21.19)	−1.870	0.062
Parity (*n*(%)) ^+^				
≥1	14 (58.3)	58 (45.3)	1.025	0.311
0	10 (41.7)	70 (54.7)		
FPG in the first trimester (mmol/L) ^#^	5.17 (4.90, 5.50)	4.71 (4.67,4.86)	−3.682	<0.001
FPG in the second trimester (mmol/L) ^#^	4.80 (4.62, 5.60)	4.55 (4.51,4.71)	−2.318	0.020
Gestational weight gain (kg) ^#^	11.50 (10.38, 14.27)	11.30 (11.53,13.15)	−0.695	0.487
OGTT 0 min (mmol/L) ^#^	4.83 (4.65, 5.34)	4.67 (4.59,4.78)	−1.433	0.152
OGTT 60 min (mmol/L) *	10.245 ± 1.261	9.547 ± 1.523	−1.988	0.049
OGTT 120 min (mmol/L) *	8.697 ± 1.650	8.109 ± 1.472	−1.664	0.098

* Data are mean ± SDs. ^#^ Data are median and interquartile ranges. ^+^ Data are numbers and %.

**Table 5 nutrients-14-00359-t005:** Comparison of dyslipidemia rate between the two groups.

Serum Lipid Parameter (mmol/L)	CRD Group ^+^ (*n* = 152)	Control Group ^+^ (*n* = 113)	χ^2^	*p*
The second trimester				
TC	38 (25.0)	28 (25.2)	0.002	0.99
TG	16 (10.5)	6 (5.4)	2.195	0.14
HDLC	40 (26.5)	29 (26.1)	0.004	0.99
LDLC	10 (6.6)	16 (14.4)	4.345	0.04
The third trimester				
TC	46 (30.5)	37 (33.0)	0.197	0.66
TG	17 (11.3)	12 (10.7)	0.019	0.99
HDLC	27 (17.9)	21 (18.8)	0.033	0.87
LDLC	20 (13.2)	18 (16.1)	0.416	0.60

^+^ Data are presented in numbers (percentage).

**Table 6 nutrients-14-00359-t006:** Comparison of nutritional status between the two groups.

Nutritional Parameter (g/L)	CRD Group (*n* = 152)	Control Group (*n* = 113)	t	*p*
The second trimester				
TP	66.483 ± 3.920	67.175 ± 4.001	1.332	0.184
ALB	36.967 ± 2.426	37.105 ± 2.615	0.421	0.674
HB	118.689 ± 9.927	117.496 ± 11.758	−0.889	0.375
The third trimester				
TP	62.715 ± 14.410	63.605 ± 12.897	0.520	0.603
ALB	35.639 ± 2.180	35.304 ± 1.897	−1.287	0.199
HB	125.684 ± 10.429	124.829 ± 11.596	−0.626	0.532

## Data Availability

Not applicable.

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
