# Peer review of "Effect of Carbohydrate-Restricted Dietary Pattern on Insulin Treatment Rate, Lipid Metabolism and Nutritional Status in Pregnant Women with Gestational Diabetes in Beijing, China"

_nutrients, 2022, doi:10.3390/nu14020359_

Round 1

Reviewer 1 Report

In the present study, Cui et al address the effect of carbohydrate-restricted dietary (CRD) pattern on glycemic management in women with gestational diabetes mellitus, using for this its effect on the proportion and time of onset of insulin treatment. The study is interesting and the topic is currently under active investigation.

In general, the article is well written, although it does present some aspects that must be improved. The comments refer particularly to minor defects, with no major defects found.

Introduction:

  • It remains to define what is understood by CRD and to include references in this regard, as well as to indicate what is currently known about the use of CRD in the management of gestational diabetes (i.e. Nutrients 2019, 11, 1737; Nutrients 2021, 13, 2599).
  • Include references about dyslipidemia in gestational diabetes and why CRD could help in its management.
  • It remains to justify why it is relevant to investigate the metabolism of calcium and phosphorus in women with gestational diabetes.

Materials and Methods

  • Justify why decided to use a carbohydrate intake of 47 E%. Include references.

Results

  • It remains to indicate what was the distribution of macronutrients in the diet of the control group.
  • Indicate the effective duration of the intervention with CRD, including measures of dispersion (s.d., range).
  • On page 8 (lines 231-233), the authors state that they used a Mann Whitney test to assess the difference in the prevalence of elevated LDL cholesterol between groups when they should have used chi square.
  • Include measurement units in figure 2 and table 6.

Reviewer 2 Report

This is an interesting paper that can contribute to GDM management in clinical practice. However, it has some issues that should be addressed as they can clarify the methodology used, as well as improve the soundness of the results: 

1) regarding the introduction section, the author did not explain why they want to evaluate calcium and phosphorous metabolism in this particular population; 

2) Although authors refer in the abstract this is a retrospective study, they use the present, the past and future tenses; as an example please see line 141.  

3) As a retrospective study, how did researchers know in such detail the dietary intervention? Are all these pregnant women follow in the same hospital? 

4) The glucometer was similar for all pregnant women included in this study or each woman select its own glucometer? 

5) Please review Table 2 formatting; I do not understand the meaning of “parity” and “≥1 or 0” classification. 

6) The authors recognized (in the introduction section) the importance of exercise in GDM management, but they do not present any information about this. In my opinion, this is a critical issue and should also be addressed if it has any effect on glycemic control and insulin use/need. 

7) Is there any data about HbA1c? This is also an important issue regarding glycemic control and can be associated with insulin use (to be included in univariate analysis, if possible). 

8) One of the most difficulties for those GDM women is to control fasting blood glucose at the morning and this is harder to be improved by nutritional therapy and/or exercise. Do the authors evaluate the influence of this parameter on insulin use? Why not include it in univariate analysis? 

Minor issues: 

Line 91 please correct the glucose units (mg/dl). 

Line 128, do you mean BMI ≥ 24 or 25 kg/m2? 

Write in full, for the first time of appearance: SMBG, FPG, HB, ALB and TP). 

Figure 2, please change “placebo” for “control group”   

Round 2

Reviewer 1 Report

Dear authors:
Thank you very much for taking into consideration the comments made in the previous review. Although most of the observations were corrected, I would like to indicate that a couple of details remain:
1. In table 4, the percentages corresponding to the number of multiparous and primiparous mothers are misreported. For comparison purposes, they should be expressed as a percentage of first-time and multiparous mothers within each group (with and without insulin), that is, the group of mothers treated with insulin presents 58.3% of multiparous mothers and 41.7% of primiparous mothers, while the group without insulin presented 45.3% and 54.7% of multiparous and primiparous mothers, respectively.
2. In caption of figure 2 it does not appear what panel A and B correspond to. Although this appears in the text, the tables should be self-explanatory.
3. Finally, and the most important, although it is true that the article is focused on the restriction of carbohydrate intake, it is necessary to know the composition of the control diet in terms of energy and macronutrients, since the magnitude of the intervention is specifically related to how much carbohydrate restricted diet is different with respect to the control group.

Author Response

Point 1:  In table 4, the percentages corresponding to the number of multiparous and primiparous mothers are misreported. For comparison purposes, they should be expressed as a percentage of first-time and multiparous mothers within each group (with and without insulin), that is, the group of mothers treated with insulin presents 58.3% of multiparous mothers and 41.7% of primiparous mothers, while the group without insulin presented 45.3% and 54.7% of multiparous and primiparous mothers, respectively.

Response 1: The relevant data in Table 4 have been revised.

Point 2: In caption of figure 2 it does not appear what panel A and B correspond to. Although this appears in the text, the tables should be self-explanatory.

Response 2: Descriptions of panel A and B have been added in caption of figure 2, and the original expression in the text below the figure has been simplified at the same time.

Point 3:  Finally, and the most important, although it is true that the article is focused on the restriction of carbohydrate intake, it is necessary to know the composition of the control diet in terms of energy and macronutrients, since the magnitude of the intervention is specifically related to how much carbohydrate restricted diet is different with respect to the control group.

Response 3: The composition of the control diet has been supplemented in the third paragraph of 2.2 Intervention in Materials and Methods.

Thank you again for your review, which benefited us a lot.

Reviewer 2 Report

I do not have additional comments or suggestions for authors.

Author Response

Thank you again for your review.